DOI: 10.1038/s41467-017-02566-1　　**OPEN**

# Defects controlled hole doping and multivalley transport in SnSe single crystals

Zhen Wang[1], Congcong Fan[2,3], Zhixuan Shen[1], Chenqiang Hua[1], Qifeng Hu[1], Feng Sheng[1], Yunhao Lu[4], Hanyan Fang[5], Zhizhan Qiu[5], Jiong Lu[5], Zhengtai Liu[2,3], Wanling Liu[2,3], Yaobo Huang [6], Zhu-An Xu [1,7,8], D.W. Shen [2,3,6] & Yi Zheng [1,7,8]

SnSe is a promising thermoelectric material with record-breaking figure of merit. However, to date a comprehensive understanding of the electronic structure and most critically, the self-hole-doping mechanism in SnSe is still absent. Here we report the highly anisotropic electronic structure of SnSe investigated by angle-resolved photoemission spectroscopy, in which a unique pudding-mould-shaped valence band with quasi-linear energy dispersion is revealed. We prove that p-type doping in SnSe is extrinsically controlled by local phase segregation of $SnSe_2$ microdomains via interfacial charge transferring. The multivalley nature of the pudding-mould band is manifested in quantum transport by crystallographic axis-dependent weak localisation and exotic non-saturating negative magnetoresistance. Strikingly, quantum oscillations also reveal 3D Fermi surface with unusual interlayer coupling strength in p-SnSe, in which individual monolayers are interwoven by peculiar point dislocation defects. Our results suggest that defect engineering may provide versatile routes in improving the thermoelectric performance of the SnSe family.

[1] Department of Physics, Zhejiang University, Hangzhou 310027, China. [2] State Key Laboratory of Functional Materials for Informatics, Shanghai Institute of Microsystem and Information Technology (SIMIT), Chinese Academy of Sciences, Shanghai 200050, China. [3] Center for Excellence in Superconducting Electronics (CENSE), Chinese Academy of Sciences, Shanghai 200050, China. [4] State Key Lab of Silicon Materials, Zhejiang University, Hangzhou 310027, China. [5] Department of Chemistry, National University of Singapore, 3 Science Drive 3, Singapore 117543, Singapore. [6] Shanghai Synchrotron Radiation Facility, Shanghai Institute of Applied Physics, Chinese Academy of Science, Shanghai 201204, China. [7] Zhejiang California International NanoSystems Institute, Zhejiang University, Hangzhou 310058, China. [8] Collaborative Innovation Centre of Advanced Microstructures, Nanjing 210093, China. Zhen Wang, Congcong Fan and Zhixuan Shen contributed equally to this work. Correspondence and requests for materials should be addressed to D.W.S. (email: dwshen@mail.sim.ac.cn) or to Y.Z. (email: phyzhengyi@zju.edu.cn)

Although the syntheses and electrical measurements of the Group IV monochalcogenides SnS, SnSe, GeS and GeSe can be dated back to 1956[1], the interests in this family soar very recently after the report of an extraordinary thermo-electric conversion efficiency of $ZT = 2.6$ at 923 K in SnSe[2]. For a given temperature ($T$), the dimensionless value of $ZT$ is known to be complex interplay between the Seebeck coefficient ($S$), the electrical conductivity ($\sigma$), and the thermal conductivity ($\kappa$), as defined by the formula of $ZT = (S^2\sigma/\kappa)T$. The unprecedented $ZT$ in SnSe has been attributed to an ultralow $\kappa$, due to a giant phonon anharmonic effect[2], which is supported by inelastic neutron scattering measurements[3]. Noticeably, there is a second-order phase transition in SnSe at ~810 K, from the low-$T$ $Pnma$ (#62) phase to the high-$T$ $Cmcm$ (#63) phase[4]. The $Cmcm$ phase has a small direct band gap of 0.46 eV, which is not sufficient to suppress thermal activation of electrons in the conduction band. With strong bipolar transport above 700 K, $\kappa$ in SnSe is dominated by the electronic contribution ($\kappa_{el}$) rather than the anharmonic phonon part ($\kappa_L$)[5]. It is thus proposed by Shi and Kioupakis[5] to introduce heavy hole doping to maximise the thermoelectric performance in SnSe, which has an upper limit of $ZT = S^2\sigma T/\kappa_{el}$. Using sodium as hole dopants, Zhao et al.[6] have observed drastic $ZT$ increase in p-SnSe. Compared to other doped thermoelectric materials[7,8], $S$ in Na-doped SnSe is unusually high[6], which is a strong indication of multiple pocket thermal transport. However, the physical origin of such multivalley transport is not clear, since theoretical calculations just suggest single-band dominated $S$ at the experimental doping level. Electron doping of SnSe by iodine[9] and bismuth[10] are less impressive with a highest $ZT$ value of 2.2[10], although theoretical models predict unrivalled thermoelectric performance in n-type SnSe[11–13].

Despite of the critical importance of hole doping, the low-lying electronic structure of SnSe, which is mainly composed of the resonantly bonded $p$-orbitals and intimately correlated to the formation of strongly anisotropic lattice anharmonicity[3], is not well studied by experiments. Remarkably, the resonant $p$-bonding network in the SnSe family forms characteristic puckering bilayer structure similar to black phosphorus (BP)[14], which is known to show distinct anisotropic physical properties along different crystallographic axes[15,16]. In SnSe, highly anisotropic thermoelectric properties, i.e., $S$, $\sigma$ and $\kappa$, have also been observed with a strong doping dependence[2,6]. However, a systematic charge transport study in complementary to the electronic structure has not been achieved yet. Most critically, the as-synthesised SnSe is moderately hole doped, instead of semiconducting with a predicted indirect band gap. Using scanning tunnelling microscopy, Duvjir et al.[17] suggest that Sn vacancies may play a decisive role in self-hole doping. A fundamental understanding of the self-hole-doping mechanism will allow the community to develop an effective doping strategy for SnSe, and reconcile the discrepancy between theoretical predictions and experiments.

In this work, we report the low-energy electronic structure of SnSe single crystals probed by complementary techniques of angle-resolved photoemission spectroscopy (ARPES) and quantum transport measurements. We elucidate the physical origins of the hole-doping mechanism and striking three-dimensionality of the Fermi surface (FS) in two peculiar types of defects, namely local phase segregation of SnSe$_2$ and point dislocations, respectively. Using high-resolution ARPES, we have resolved for the first time the unique pudding-mould-shaped valence band (VB) of p-SnSe, characterised by two equivalent VB maximums in close proximity along the high-symmetry $\Gamma$–$Z$ line and quasi-linear energy dispersion away from the relatively flat band tops. We show unambiguous evidences that locally segregated SnSe$_2$ microdomains determine the observed hole doping in SnSe via

interfacial charge transferring. By utilising different growth methods and fine-tuning parameters, we demonstrate that the extrinsic hole doping in SnSe can be widely tuned from semi-conducting to $1.23 \times 10^{18}$ cm$^{-3}$. The ARPES deduced non-parabolic band parameters and multivalley Fermi surface are well supported by $T$-dependent quantum transport measurements, which show distinctive non-saturating negative magne-toresistance (NMR) and highly anisotropic weak localisation. These exotic quantum phenomena are coincident with the puckering $c$ axis of SnSe and show notable hole-doping dependence, suggesting the importance of strong intervalley scattering assisted by in-plane ferroelectric dipole field. Strikingly, quantum oscillations also reveal unusual interlayer coupling strength and three-dimensional (3D) FS in p-SnSe, contradicting to the two-dimensional (2D) nature of the BP-type layered structure. Using atomic force microscopy, it is found that SnSe monolayers (MLs) are heavily interwoven by high density of point dislocations, which retain the anti-ferroelectric stacking order in the bulk but interconnect two second-nearest-neighbouring MLs with the same dipole orientation by forming point defects.

## Results

**Single-crystal growth and characterisation.** Figure 1a illustrates the layered structure of SnSe, which is essentially a binary ana-logue of BP[14] with alternating Sn (red) and Se (blue) rows forming puckered bilayer structure in the $b$–$c$ plane. Like BP, the two principal crystallographic axes of $b$ and $c$ represent the 'zig-zag' and 'armchair' directions[15], respectively. However, due to the high polarizability of the binary lattice, the SnSe family forms distinct anti-ferroelectric stacking order along the interlayer $a$ axis, when two neighbouring MLs reverse the in-plane dipole orientation along the $c$ axis[18,19]. As summarised in Table 1, we have synthesised 12 batches of SnSe single crystals using three growth methods, including self-flux (SF), Bridgeman (BR) and physical vapour deposition (PVD), combined with various growth parameters (see 'Methods' section, Supplementary Figure 1 and Supplementary Note 1 for the details). For stoichio-metrically synthesised SnSe, vacancies are very rarely seen under qPlus non-contact atomic force microscopy (nc-AFM), see Fig. 1b. The high quality of our single crystals are also confirmed by powder X-ray diffraction (XRD) of ground polycrystal sam-ples, showing sharp Bragg diffraction peaks for the #62 space group (Supplementary Figures 2 and 3), and by Raman spectro-scopy, with four fingerprinting phonon modes of $A_g^1$, $B_g^3$, $A_g^2$ and $A_g^3$ for the $Pnma$ phase[20,21] (Supplementary Figures 4 and 5). Noticeably, SnSe crystals tend to form large edges along the diagonal axis of the $b$–$c$ plane. It is thus crucial to determine the crystallographic orientations of individual samples by XRD before physical characterisations and transport measurements (see Supplementary Figure 3 and Supplementary Note 2). Typical physical dimensions for sliced rectangular SnSe single crystals with shining surface are $a = 1$ mm, $b = 6$ mm and $c = 3$ mm, respectively (Supplementary Figure 2).

Remarkably, without the introduction of any external dopants, different batches of SnSe single crystals exhibit distinct conduc-tion behaviours, ranging from semiconductivity to p-type degenerate semiconductors. In a brief summary, the flux cooling rates from the highest growth temperature (950 °C) determine the hole-doping levels and thus the resistivity of SnSe. For the SF and BR methods, fast cooling in 24 h to 400 °C (SF1, SF6, SF7, SF11, SF12, BR1 and BR2) leads to metallic characteristics, while much slower cooling in 7 days to 400 °C (SF8) produces semiconducting SnSe with thermal activation behaviour. Using optical microscopy and ambient multi-mode AFM, we have identified two unique types of defects in SnSe, namely point dislocations and SnSe$_2$

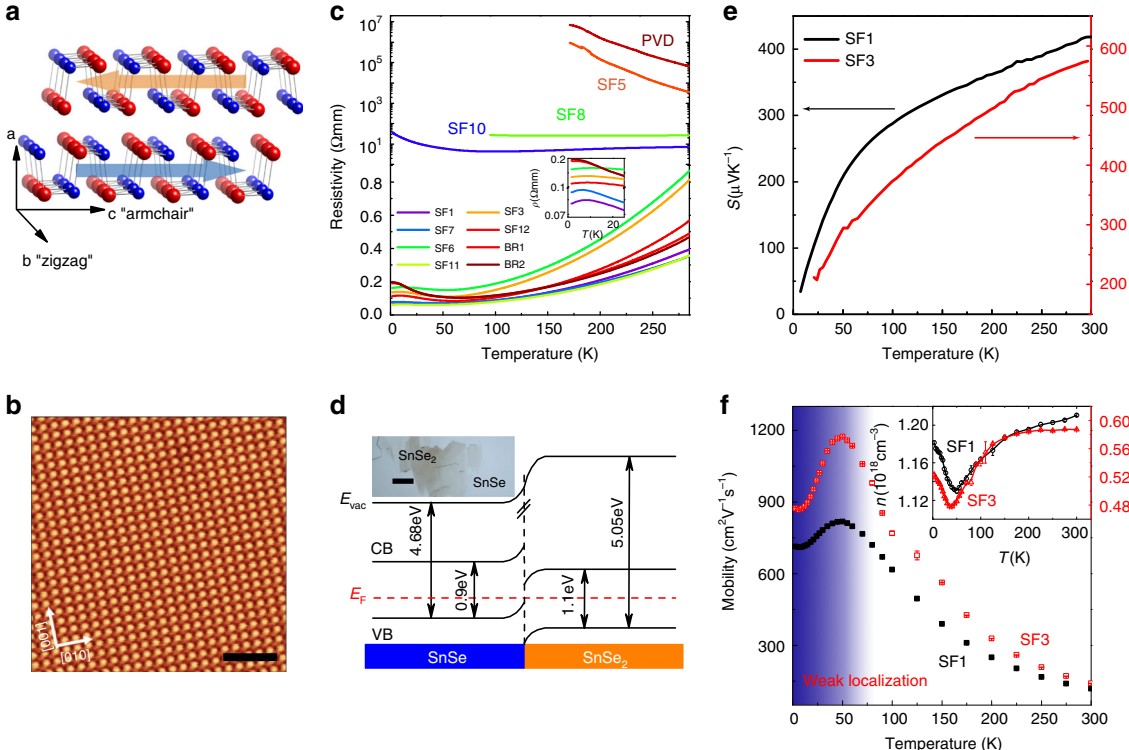

**Fig. 1** Characterisation of SnSe single crystals. **a** SnSe crystallises into the BP-type puckered bilayer structure, by forming alternating Sn- and Se-rows along the *b*-axis. The binary lattice is highly polarised with a net ferroelectric dipole field aligned with the *c*-axis. Here the red and blue balls represent Sn and Se atoms respectively. **b** Atomic-resolution nc-AFM image of stoichiometric SF1-SnSe, showing no existence of vacancies. The scale bar is 2 nm. **c** *T*-dependent $\rho$ curves for all batches of crystals, showing various conduction behaviour from metallic (SF1, SF3, SF6, SF7, SF11, SF12, BR1, and BR2) to semiconducting (SF5, SF8, SF10, and PVD). Below 50 K, all metallic batches exhibit upturns in $\rho$ vs $T$ curves, a manifestation of WL. The inset zooms in the $\rho$ vs $T$ curves below 30 K. **d** Energy level alignment of SnSe/SnSe₂ heterostructure. Interfacial charge transfer leads to p-doping in SnSe. Note that the exact value of $E_F$ depends on both the concentration and thickness of SnSe₂ microdomains. The inset is a typical optical image of SnSe₂ microdomains on SnSe. The scale bar is 10 μm. **e** High $S_b$ of SF1- and SF3-SnSe, which are in agreement with the literature reports. **f** *T*-dependent Hall measurements of SF1- and SF3-SnSe reveal mobility enhancement at low *T*, which is typical metallic behaviour. Note that the mobility drops below 50 K is due to WL

### Table 1 Physical properties of SnSe single crystals grown by three different methods

| Batches | Flux cooling rate | SnSe₂ ratio[a] | Se:Sn | $n$ ($10^{18}$ cm⁻³) | | Resistivity ($\Omega$ mm) | |
|---|---|---|---|---|---|---|---|
| | | | | SdH[b] | Hall[RT] | 2 K | 275 K |
| SF1 | Fast cooling | 0.9% | 1 | 0.825 | 1.16 | 0.0746 | 0.360 |
| SF3 | 3 days to 673 K | <0.3% | 1 | 0.502 | 0.592 | 0.136 | 0.757 |
| SF6 | Fast cooling | 0.3% | 1 | 0.704 | 0.69 | 0.162 | 0.812 |
| SF7 | Fast cooling | 3.73% | 1 | 0.836 | 1.23 | 0.0776 | 0.333 |
| SF11 | Fast cooling | 2.6% | 1 | 0.480 | 1.079 | 0.0604 | 0.333 |
| SF12 | Fast cooling | <0.3% | 0.95 | 0.472 | 0.794 | 0.191 | 0.457 |
| BR1 | Fast cooling | 2.49% | 1 | 0.423 | 0.699 | 0.113 | 0.525 |
| BR2[c] | Fast cooling | 0.5–1.5% | 1 | 0.395 | 0.591 | 0.198 | 0.436 |
| SF8 | 7 days to 673 K | <0.1% | 1 | — | — | — | 26.4 |
| SF5[d] | 3 days to 673 K | None | 1 | — | — | — | 8303 |
| SF10[d] | Fast cooling | <0.1% | 1 | — | — | 36.74 | 6.93 |
| PVD | — | None | 1 | — | — | — | 137760 |

[a] Coverage of SnSe₂ after exfoliating surface layers. Statistics by ×500 optical microscope
[b] $n_{sdh}$ is calculated by $\frac{2k_F^3}{3\pi^2}$, assuming spherical FSs. A factor of two is multiplied to reflect the double valley transport
[c] Crystal dependent. SnSe₂ microdomains thinner than 5 MLs are not distinguishable under optical microscopy
[d] Sn and Se are ground and thoroughly mixed before the growth

microdomains embedded in SnSe single crystals, respectively (Supplementary Figure 6 and Supplementary Note 3). Unexpectedly, there is a strong correlation between the conduction behaviour in SnSe and the concentrations of SnSe₂ microcrystals, which are typically several tens micrometres in lateral sizes and a few MLs to several tens MLs in thickness (Supplementary Figure 6). With a threshold of ~0.3% SnSe₂, the resistivity vs temperature ($\rho - T$) curves of SnSe becomes metallic over the whole *T* range from room temperature (RT; 300 K) to 1.5 K, while semiconducting SnSe shows negligible concentration of SnSe₂

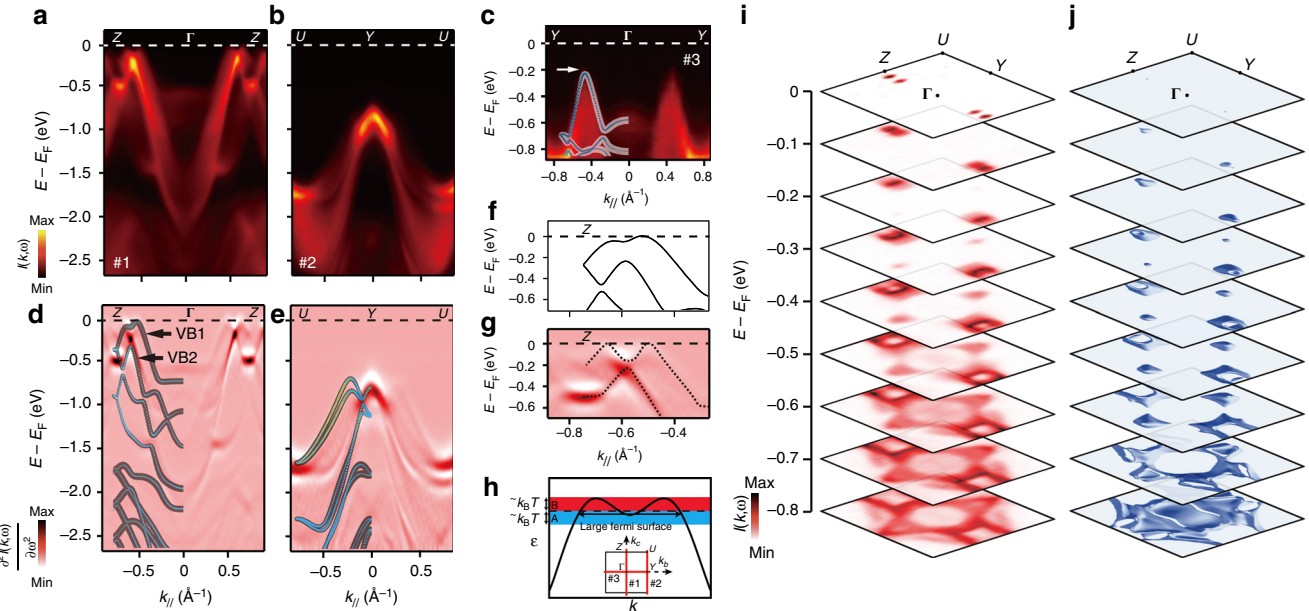

**Fig. 2** ARPES resolved electronic structure of p-SnSe. ARPES measures band dispersions along high-symmetry directions of Z–Γ–Z **a**, U–Y–U **b**, and Y–Γ–Y **c**, taken with 50 eV photon energy. The second derivative plots **d** and **e**, corresponding to **a** and **b**, respectively, are directly compared with the DFT calculations, in which the weight of Sn 5s and Se 4p orbitals are represented by light blue and yellow colours, respectively. A close-up of the top VBs along Γ – Z reveals pronounced differences between theoretical band dispersion **f** and the ARPES measured results **g**, as highlighted by the X-shaped dashed lines. **h** A schematic plot of a pudding-mould shaped VB with corrugations, which leads to giant S due to the band geometry effect. The inset of **h** indicates the ARPES cut directions in the projected two-dimensional first Brillouin zone (BZ). **i, j** Comparison of stacked plots of constant energy contours at different $E_B$, which show good agreement between ARPES and DFT when $E_B > 0.2$ eV. Black squares represent the boundary of the first BZ

microdomains (Fig. 1c). The formation of $SnSe_2$ microdomains can be explained by local phase segregation in Se-rich areas. To test the hypothesis, we have synthesised two extra batches of SnSe crystals, by grinding stoichiometric SnSe powder in argon atmosphere for 1 h before the growth. As shown in Table 1, the grinding step effectively suppresses inhomogeneity in the growth flux, and no concentration of $SnSe_2$ is observed in the slow cooling batch of SF5. For the fast cooling SF10 batch, very-low ratio of $SnSe_2$ can still be found, which is not surprising since $SnSe_2$ nucleations (~650 °C) may start during the late stage of crystal growth. We have further verified the claim by the PVD method, in which SF1-type single crystals containing ~1% $SnSe_2$ are ground into polycrystal powder and used as the PVD source. In a temperature gradient of 800–700 °C over 12 cm, a pure phase of SnSe has been obtained by the sublimation and re-deposition process, while $SnSe_2$ was removed due to the low melting point.

The hole-doping mechanism in SnSe bulk by intercalated $SnSe_2$ microdomains can be well understood by interfacial charge transfer in the framework of Anderson's rule[22]. As sketched in Fig. 1d, the energy band diagrams of $SnSe/SnSe_2$ semiconductor heterostructure are formed by vacuum level continuity and Fermi level ($E_F$) alignment at the interface. Density functional theory (DFT) calculations allow us to determine the VB maximums of bulk SnSe and $SnSe_2$, which are 4.68 and 5.05 eV, respectively. Such large VB energy difference leads to interfacial charge transfer from SnSe to $SnSe_2$, and thus, an effective hole doping in the SnSe bulk. This charge transfer–controlled hole-doping mechanism is verified by scanning Kelvin probe microscopy, which directly measures the surface $E_F$ of p-SnSe samples with an energy resolution better than 50 meV. For BR1-SnSe, the deduced $E_F$ for bare SnSe surface and few-layer $SnSe_2$/SnSe heterostructure on the same sample are 4.42 and 4.58 eV, respectively (Supplementary Figure 7). Compared to intrinsic semiconducting SnSe ($E_F = 4.23$ eV), the Fermi energy of BR1-SnSe crystals is shifted down by 0.19 eV, corresponding to p-type doping in the

BR1-SnSe bulk. For few-layer $SnSe_2$/SnSe heterostructure, the Fermi energy (4.58 eV) is shifted up from the bulk value of 4.85 eV, providing an unambiguous evidence of electron transfer-in from SnSe (Supplementary Figure 7).

Intriguingly, we have noticed that SnSe single crystals with ultrahigh ZT values in the literatures all belong to the metallic type in our study. In particularly, the RT hole doping of the SF3 batch (~$6 \times 10^{17}$ cm$^{-3}$) closely matches the crystals in the reference report prepared by the BR method[2], which holds the highest record of ZT value. As shown in Fig. 1e, the experimental RT $S_b$ of SF3-SnSe (~570 μVK$^{-1}$) also shows striking consistency with the report in ref. [2]. Our results strongly suggest the critical importance of the flux cooling rate in determining the thermo-electric transport properties of SnSe, by controlling the concentration of $SnSe_2$ microdomains in the bulk. This is mainly due to the fact that the nucleation of SnSe and $SnSe_2$ are rather close in temperatures ~820 °C and ~650 °C, respectively. For SnSe crystal with very-fast flux cooling rates, Sn vacancies and Se interstitials, which may effectively scatter lattice phonons, have been imaged by scanning transmission electron microscope[23]. However, these two specific types of defects are not present in our SnSe single crystals.

As a comparison, we have also systematically studied the T-dependent $S_b$ of the SF1 batch, which have two times higher hole doping of $1.23 \times 10^{18}$ cm$^{-3}$. As shown by the black solid line in Fig. 1e, hole doping slightly reduces RT $S_b$ to ~420 μVK$^{-1}$, in good agreements with both the theoretical calculations[5] and experiments[6,24]. By cooling down the SF3 samples to 1.5 K, we have observed that the ρ–T curves show a minimum at around 50 K, followed by an upturn down to the base temperature. Using Hall measurements (Supplementary Figure 8 and Supplementary Note 4), we determined that the ρ–T behaviour from RT to 50 K is mainly due to hole mobility enhancement at low temperatures (Fig. 1f and Supplementary Figures 9–11), which is typical for metals, but not consistent with thermally activated charge

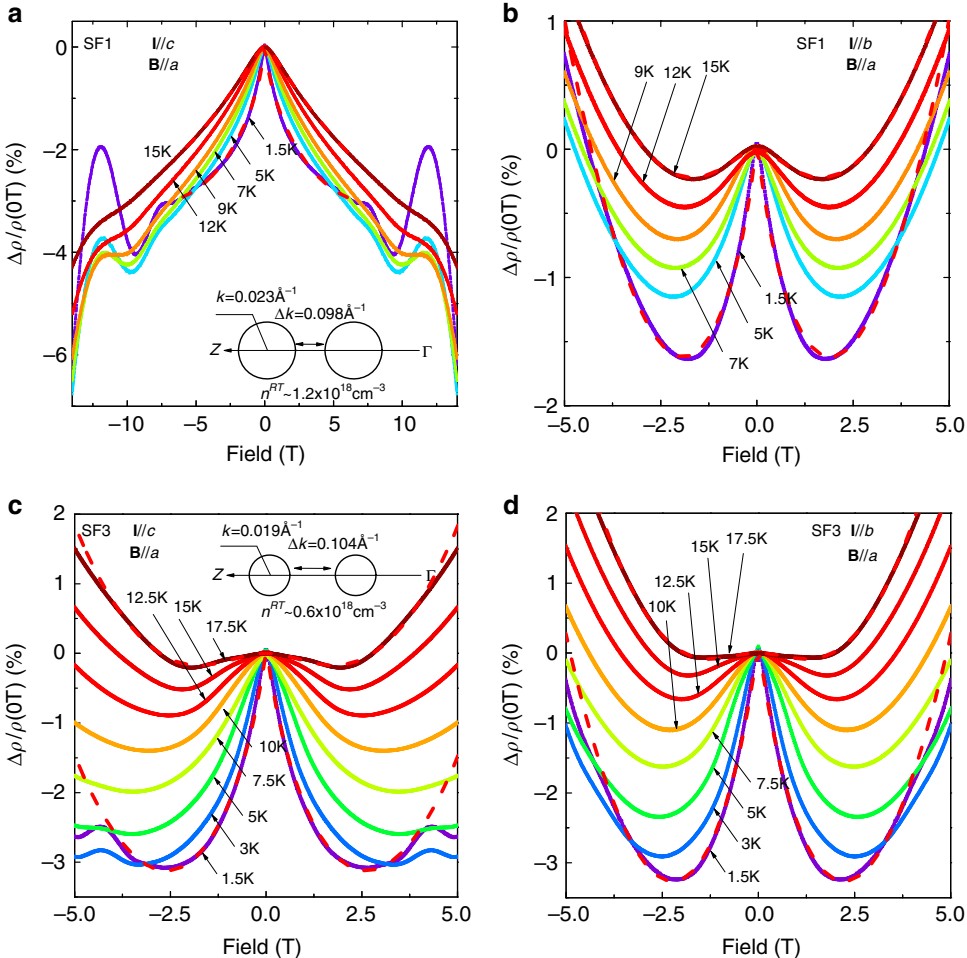

**Fig. 3** Doping-dependent highly anisotropic weak localisation in p-SnSe. **a, b** NMR in SF1-SnSe for **I**∥c and **I**∥b, respectively, when the positive WL correction to resistivity is suppressed by external field. The red-dash lines are the fitting results using the WL model in the main text, which explains the evolution of low-field NMR very well by $T$-dependent $l_\phi$. Strikingly, the NMR growth of SF1-SnSe for **I**∥c is non-saturating up to 14 T, while NMR for **I**∥b dwindles as a function of $T$ and is only dominant below 2 T. The low-field NMR in SF3-SnSe becomes less anisotropic, however, the overall WL correction to resistivity is stronger in these samples (see **c, d**). The insets of **a, c** show the sketches of the FS cross-sections for SF1- and SF3-SnSe, respectively. For **I**∥c, intervalley scattering assisted by the in-plane dipole field along the c-axis gets enhanced, when the two pudding-mould valleys become closer by p-doping. Note that the SnSe₂ phase segregation may lead to non-uniform doping in p-SnSe, making the two pudding-mould valleys even closer in localised areas

transport in semiconductors. Such metallic behaviour is general for different batches of p-SnSe with significant presence of SnSe₂ microdomains. As shown in Fig. 1f, hole mobility of the SF1 sample shows the same $T$ dependence as SF3-SnSe, despite that the highest mobility point at 50 K is 1.5 times larger in the latter.

**Electronic structure of p-SnSe.** To understand the metallic behaviour in p-SnSe single crystals and the corresponding correlation with their high $ZT$ values, we performed systematic ARPES measurements to investigate their band structure (see 'Methods' section for details). Figure 2a–h summarise the direct comparison of experimental band dispersions and DFT calculations along three high-symmetry directions of $Z–\Gamma–Z$, $U–Y–U$ and $Y–\Gamma–Y$, respectively. Overall, as a weakly correlated system, the ARPES results of SnSe VBs can be well reproduced by DFT after a rigid shift of chemical potential, indicating that randomly intercalated SnSe₂ microdomains just play the role of charge reservoir but do not affect the electronic structure of SnSe significantly, which is consistent with the aforementioned interfacial charge transfer model. This argument could be further confirmed

by the close resemblance between the experimental and DFT constant energy contour plots (Fig. 2i–j). As shown in Fig. 2a–d, ARPES measurements reveal two side-by-side ellipsoid hole pockets along $\Gamma–Z$ in the vicinity of $E_F$, which is in good agreement with the metallic transport properties of these samples. By projecting the Sn and Se orbitals to the DFT-calculated VBs, we find that this highest VB (VB1) is dominated by Se $4p$ states, with a relatively low weight of the Sn $5s$ orbital (Fig. 2d). However, the detailed band structure in the vicinity of the $E_F$ of SnSe still shows some pronounced differences between the DFT and ARPES results. For DFT, as highlighted in Fig. 2f, the band top of VB1 is found along $\Gamma–Z$ with the Brillouin zone positions of (0.00, 0.00 and 0.52). Moving towards the zone boundary, there is another local VB maximum located at (0.00, 0.00 and 0.64), which is yet about 50 meV lower than the band top. However, the two ARPES determined VB1 maxima are nearly equal in energy, leading to the formation of the unique pudding-mould-shaped band[25] in Fig. 2g.

As illustrated in Fig. 2h, a pudding-mould band is characterised by a relatively flat portion at the top which bends sharply to a highly dispersive lower part. Consequently, when the doping level is moderately tuned to be around the bending

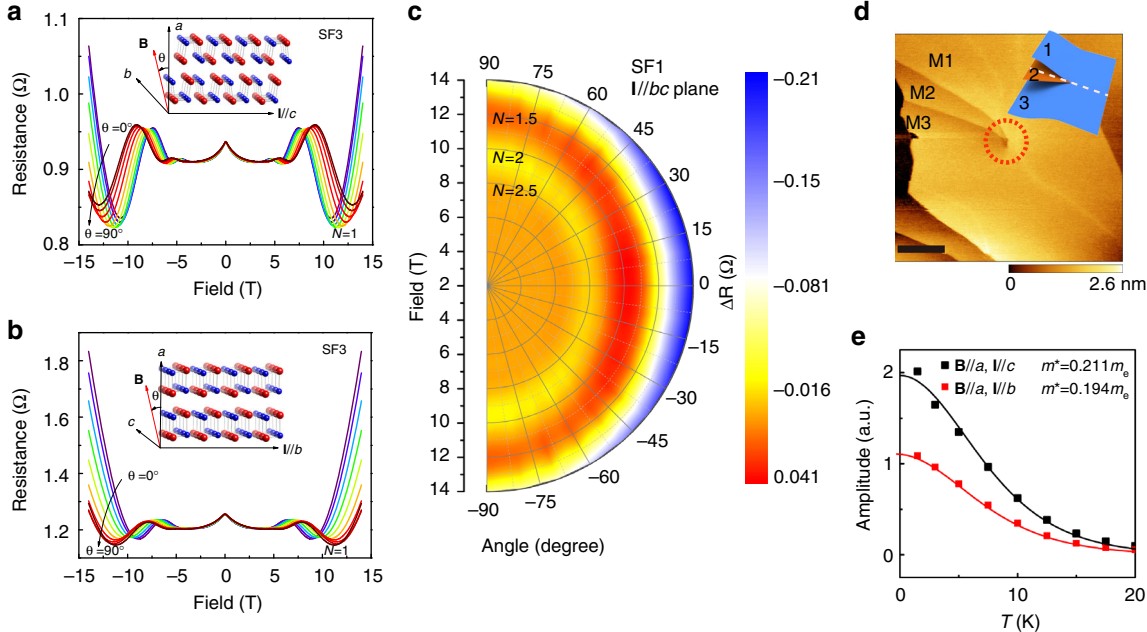

**Fig. 4** 3D Fermi surface in p-SnSe induced by point dislocations. **a, b** Angle-dependent SdH oscillations of SF3-SnSe at 1.5 K for **I**||c and **I**||b, respectively. The insets explain the measurement configuration, in which transverse **B** is rotated from the a-axis to the corresponding in-plane direction. **c** Polar contour plot of the θ-dependent SdH oscillations in SF1-SnSe for **I**||bc. The corresponding 3D FS is a elongated ellipsoid in the b–c plane. **d** Typical ambient AFM image of a point dislocation in SnSe crystals, showing the interconnection between the top ML and the third ML. The inset illustrates the reversing of stacking sequence between the top and second ML when crossing the point defect along the white dashed line. The scale bar is 200 nm. **e** LK-model fit of the T-dependent SdH amplitudes for SF3-SnSe, yielding similar effective mass of around $0.2m_e$ for both **I** directions. The out-of-plane $m^*$ measured by in-plane transverse **B** is only slightly higher than these in-plane values

regime, S would become unusually large due to a rapid change in charge carrier mobility across $E_F$[25]. In the case of SF1-SnSe presented here, such band geometry effect could be more drastically enhanced by the quasi-linear band dispersions of VB1, as highlighted by the X-shaped dashed lines in Fig. 2g. Such X-shaped VB1 also explains the experimental observation of relatively high electrical conductivity in p-SnSe, as a result of small effective mass ($m^*$) and multivalley contributions to charge transport. By plotting constant energy contours, we can evaluate the FS evolution as a function of rigid chemical potential shifts ($E_B$), which is equivalent to hole doping changes, and thus get insights into the thermoelectric performance of heavily p-doped SnSe. As shown in Fig. 2i–j, multiple sets of VBs can be activated in charge transport by gradually increasing hole doping. These include another local maximum of VB1 along Γ−Y and the VB2 valley for $E_B \geqslant 0.2$ eV, and a giant flat band at the zone centre when $E_B$ reaches ~0.6 eV. For the highest experimental hole doping of $4 \times 10^{19}$ cm$^{-3}$ [6], we can estimate a $E_B$ of 0.13 eV by assuming a constant effective mass of $m^* = 0.2m_e$, which is still distant from the saddle point (0.18 eV) of the pudding-mould VB1. It may thus conclude that the observed usually high S for Na-doped SnSe is a combination of band geometry effect and multivalley thermal transport (Supplementary Figure 12 and Supplementary Note 5).

**Quantum localisation in p-SnSe**. The resistivity upturn of p-SnSe below 50 K is a transport signature of weak localisation (WL), which is evident by NMR in low fields when constructive quantum interference is suppressed by breaking time reversal symmetry[26]. For p-SnSe in this study, the doping levels are well below $2 \times 10^{18}$ cm$^{-3}$, corresponding to $E_F$ which is located quite close the two VB1 maxima. Thus, the charge transport in our samples are also dominated by the multivalley FSs of the

pudding-mould shaped VB1, which produces exotic quantum phenomena in p-SnSe. As shown in Fig. 3a, when the current flow (**I**) is driven along the c-axis, increasing transverse field (**B**||a-axis) gradually weakens the positive WL correction to resistivity, manifested by pronounced NMR at low fields. Strikingly, the NMR growth in SF1-SnSe does not show any sign of saturation up to the maximum field of 14 T (see also Supplementary Figure 13 and Supplementary Note 6 for BR1), and becomes more conspicuous at elevated T above 10 K, when Shubnikov-de Haas (SdH) oscillations are effectively suppressed (Fig. 3a). Such exotic behaviour for **I**||c are in stark contrast to the MR characteristics for **I**||b, in which NMR magnitudes dwindle as a function of T and are only dominant at low fields below 2 T before normal positive MR prevails (Fig. 3b). The low-field NMR in SF3-SnSe becomes less anisotropic on the crystallographic axis dependence, while the overall WL correction to resistivity is stronger in these samples. As shown in Fig. 3c, with the same configurations of **I**||c and **B**||a, SF3-SnSe shows significantly larger low-filed NMR, which reaches a maximum $\Delta\rho/\rho$ change of about −3% at 2 T, compared to −2% in SF1-SnSe with two times higher p-doping. For **I**||b, NMR in SF3-SnSe at 2 T (−3.2%) is comparable to the c axis, and is larger than the b axis of SF1-SnSe (−1.7%).

For the multivalley FSs in p-SnSe, which are geometrically aligned along the Γ−Z direction, the ferroelectric dipole field along the c-axis may cause strong intervalley scattering to induce anisotropic WL phenomena. Using the ARPES results, we have compared the FSs of SF1- and SF3-SnSe (Supplementary Figure 14 and Supplementary Note 7). As show in the insets of Fig. 3a and Fig. 3c, with two times higher hole doping in SF1-SnSe, the Fermi energy is down shifted by ~5 meV, which reduces the separations between the two pudding-mould valleys from 0.104 to 0.098 Å$^{-1}$. Consequently, intervalley scattering is expected to be stronger in the case of SF1-SnSe, when the momentum mismatch Δ**p** is compensated by the dipole field acceleration of hole carriers in

the case of $\mathbf{I} \| c$. For non-relativistic fermions, an increase in intervalley scattering generally leads to the suppression of WL, due to the interruption of backscattering loops. Quantitatively, we can use a simplified model[27] to incorporate the intervalley scattering effect into WL: $\frac{\Delta\rho}{\rho^2} = -\alpha \frac{e^2}{2\pi^2\hbar} \sqrt{\frac{eB}{\hbar}} F\left(\frac{4eBl_\phi^2}{\hbar}\right) + \gamma B^2$, in which $\alpha \leqslant 1$ is a fitting coefficient representing the overall strength of WL, $B$ is the magnitude of magnetic field, $l_\phi$ is the phase coherence length, $F(x)$ is the Hurwitz zeta function, and $\gamma B^2$ is the normal quadratic MR (see Supplementary Notes 8 and 9). By fitting the $T$-dependent WL in SF1-SnSe and SF3-SnSe samples, we consistently get $\alpha_c(\text{SF1})$ less than $\alpha_b(\text{SF1})$ and $\alpha_c(\text{SF1})$ less than $\alpha_c(\text{SF3})$ from 1.5 K to 20 K (Supplementary Figures 15–17), supporting the claim of WL suppression by the highly anisotropic intervalley scattering between the two pudding-mould valleys. It is fascinating that for SF1-SnSe, the finite $\gamma B^2$ term at 1.5 K approaches to zero quickly above 5 K for $\mathbf{I} \| c$. This could also be related to the anisotropic intervalley scattering in p-SnSe, when the conventional quadratic MR due to the cyclotron movement of charge carriers became suppressed by the dipole field.

**3D Fermi surface of p-SnSe.** We further determined the FS morphology of SF1- and SF3-SnSe using angle-dependent SdH oscillations by rotating the samples in transverse $\mathbf{B}$ with two current geometries of $\mathbf{I} \| c$ and $\mathbf{I} \| b$. Unexpectedly, we have resolved ellipsoid FSs with moderate anisotropy for both batches of samples, as evident by the angular ($\theta$) evolution of the SdH peaks at 1.5 K. By fixing $\mathbf{B} \| a$ ($\theta = 0$), SdH oscillations mainly compare the anisotropy in charge carrier mobility along the $c$- and $b$-axes, respectively. As shown in Fig. 4a, b, SdH oscillations in SF3-SnSe quickly reach the quantum limit (Landau level $N = 1$) at 8.8 T for $\mathbf{I} \| b$, while 10.8 T is required for $\mathbf{I} \| c$. The observed anisotropic quantisation for $\mathbf{I} \| c$ and $\mathbf{I} \| b$ is rooted in the puckering lattice of SnSe, which has different mobility for the $b$- and $c$-axes in p-doped crystals[5]. To our surprise, the $\theta$ dependence of SdH oscillations for both current configurations are rather weaker when the transverse field is rotated from the out-of-plane ($\mathbf{B} \| a$) to the in-plane direction ($\mathbf{B} \| b$ or $\mathbf{B} \| c$). In Fig. 3c, we plotted the polar contour of the $\theta$-dependent SdH oscillations for SF1-SnSe with $\mathbf{I} \| bc$ (diagonal axis of the $b$-$c$ plane), which also reveals a 3D FS independent of current flow. These experimental results in p-SnSe can be directly compared to the 2D quantum oscillations in BP[14,28,29], which shares the same puckering bilayer structure of resonantly bonded $p$-orbitals. The absence of a $1/\cos(\theta)$ dependence in the SdH oscillation frequencies (Supplementary Figures 10, 15 and 16) and the robust quantum oscillations for in-plane $\mathbf{B}$ strongly suggest that p-SnSe has considerable interlayer coupling to suppress the 2D nature of the BP-type lattice, which is theoretically predicted to have very weak interlayer coupling energy, marginally higher than graphite[19].

Although $SnSe_2$ microcrystals play a critical role in determining the resistivity of p-SnSe, it is unlikely that the unusually enhanced interlayer coupling in SnSe is correlated to the local phase segregation. This is straightforward to see since 2D $SnSe_2$ microdomains are randomly intercalating between the puckering SnSe MLs, which basically destroys the stacking order of SnSe crystals. Instead, we found that point dislocations are highly likely to be responsible for the strong interlayer coupling and 3D FS in SnSe. As shown in Fig. 4d, the formation of point dislocations is due to the interconnection between SnSe MLs with the same even/odd sequence. This type of unique interlayer connection by a point defect is the direct consequence of anti-ferroelectric stacking between neighbouring SnSe MLs, which is thermodynamically favoured by the bulk phase. As schemed in Fig. 1a, in the *Pnma* phase, neighbouring SnSe MLs have opposite in-plane

ferroelectric dipole orientation, which prohibit the interconnection of even-odd MLs without destroying the anti-ferroelectric stacking order. On the other hand, point defects connecting the second-nearest neighbouring MLs with the same dipole orientation retain the anti-ferroelectric bulk phase by reversing the stacking sequence in the vicinity of a point dislocation, as indicated by the white dashed line in the inset of Fig. 4d. For SF1-SnSe, the dislocation density is larger than $4.5 \times 10^4$ mm$^{-2}$, corresponding to more than one dislocation site in a randomly searched area of $10 \times 10$ μm$^2$ (see Supplementary Figures 18, 19 and Note 10 for AFM statistics). Such a high density of interlayer defects interweave SnSe MLs along the $a$-axis, and thus greatly weaken the two-dimensionality of SnSe. The three-dimensionality of FS induced by point dislocations is also manifested in $m^*$, which is a quantitative parameter representing the band dispersion along different crystallographic axes. By fitting the $T$-dependent SdH amplitudes using the Lifshitz-Kosevich (LK) formula, we can get the effective mass for different $\mathbf{I}$ and $\mathbf{B}$ configurations (Fig. 4e). As summarised in Supplementary Table 2, $m^*$ deduced from quantum oscillations are comparable between the interlayer direction and the in-plane axes. Such band parameters are well supported by the ARPES results, which yield an effective mass of $m^* = 0.258 m_e$ for the pudding-mould shaped VB.

## Discussion

We have used synchrotron radiation based ARPES and quantum transport measurements to probe the electronic structure of SnSe. The first-time resolved pudding-mould VB with quasi-linear energy dispersion may hold the key to understand the extraordinary thermoelectric performance of p-SnSe. Within the experimental hole-doping levels achieved to date, charge transport in p-SnSe is contributed by two pudding-mould valleys (four hole pockets in total), which could explain the multivalley nature of the observed unusual thermoelectric transport in Na-doped SnSe. Equally important, the ultrahigh $S$ in p-SnSe also benefits from the band geometry of the pudding-mould VB, which causes drastic changes in charge carrier mobility across $E_F$. Interestingly, by further increasing hole doping to $E_B = 0.18$ eV, we will reach the saddle point of the pudding-mould VB, where thermoelectric coefficients may have extremes. It is also tempting to explore $S$ in p-SnSe at even higher $E_B$ of 0.2 eV and 0.6 eV, where multiple sets of extra valleys will be activated in thermoelectric transport, respectively. However, this would require micro-mechanical exfoliation and device fabrications so that thickness-dependent study of SnSe becomes feasible. By introducing heavy ionic-liquid doping in SnSe thin-film devices, the ultimate thermoelectric performance of both n-type and p-type SnSe could be systematically explored. In the 2D limit, such SnSe electronic devices would also be fascinating platforms to explore exotic physical phenomena, such as quantum Hall effects[29] and multiferroelectric properties[18,19].

The origin of the pudding-mould VB, which significantly deviates from the DFT calculations (see Supplementary Figure 20 and Supplementary Note 11 for the tight-binding calculations), is beyond the scope of this study. Nevertheless, it may be due to the hybridisation of the Se $p$-orbitals and Sn 5$s$ bands, which is extremely sensitive to the local SnSe polyhedron environments[3]. Indeed, we have observed that VB1 along $\Gamma - Y$, which is contributed by Se 4$p_y$ orbitals with negligible weight of Sn 5$s$ states, shows high degree of consistency between the ARPES results and DFT calculations (Fig. 2c). For the modelling of the pudding-mould VB, it is also important to take into account the unique point dislocation defects, which have drastically changed the 2D nature of ideal SnSe lattice as evident by the quantum oscillation

results. In the vicinity of these point defects, the local Sn-Se bonding network are likely to be disturbed by the reversing of stacking sequences between neighbouring MLs, leading to pronounced changes in the hybridisation of the Se 4$p$ and Sn 5$s$ orbitals. Such local strain field created by point dislocations in-principle can be well understood by nano-mechanical simulations[30–32].

Our results provide insights into the hole-doping mechanism in SnSe, which is extrinsically controlled by randomly distributed SnSe$_2$ microdomains intercalating the bulk. An alternative strategy to introduce self-hole doping in SnSe without the formation of intercalating SnSe$_2$ is vacancy engineering. As shown in Table 1, by using non-stoichiometric growth flux with insufficient Se, SnSe single crystals (SF12) with RT conduction and hole concentration comparable to stoichiometric SF11 have been prepared, which suggests strong p-doping effect by Se vacancies. A systematic transport and complementary qPlus nc-AFM study on Se-vacancy doped p-SnSe is now under way, which may clarify the correlation between vacancies and hole doping in SnSe. The highly anisotropic quantum localisation phenomena in p-SnSe also remind us that beyond the extraordinary thermoelectric performance, SnSe offers rich physics to be explored, which are rooted in the binary BP-type puckering structure.

Note added. We have noticed that there are two other ARPES papers on p-SnSe published very recently[33,34], during the revision of our work.

## Methods

**Single-crystal synthesis**. We synthesise SnSe crystals using Sn and Se powder from Alfa Aesar (99.999%), which are stored and mixed in a glove box with argon atmosphere according to the mole ratio 1:1 or 1:0.95 for different batches (details shown in Supplementary Table 1). The mixture is then sealed in an evacuated quartz tube with argon pressure below $7.5 \times 10^{-5}$ Torr. In the final step, SnSe single crystals are grown in a tubular furnace (SF) or two-temperature zone furnaces (BR and PVD), using the $T$ curves described in Supplementary Table 1. As shown in Supplementary Figure 1, for the SF method, the growth flux is placed in the highest $T$ zone in the furnace; for BR, the tilted tube end holding the flux is placed in the low $T$ zone, while the other end is placed in the high temperature zone to form a $T$ gradient for the growth section; for the PVD method, we synthesise SnSe crystals by the fast cooling SF method first. The obtained single crystals with ~1% SnSe$_2$ are ground thoroughly in glove box and then sealed again. During the PVD process, the tube is placed in a two-temperature zone furnace with a $T$ gradient of 50 K.

**Angle-resolved photoemission spectroscopy measurements**. ARPES measurements are performed at the 'Dreamline' beamline of the Shanghai Synchrotron Radiation Facility equipped with a Scienta D80 analyser. The energy resolution was set to 10 meV and the angular resolution was set to 0.2°. The samples are cleaved in situ and then measured at 13 K in a vacuum better than $5 \times 10^{-11}$ Torr. The ARPES data are collected using linearly horizontal-polarised lights with a vertical analyser slit.

**X-ray diffraction**. The crystallographic orientations and lattice structure of SnSe are determined by X-ray Bragg diffractions using a PAN-alytical X'Pert MRD diffractometer equipped with Cu $K_\alpha$ radiation and a graphite monochromator.

**Temperature-dependent transport measurements**. Rectangularly shaped SnSe samples with typical dimensions of $2 \times 1 \times 0.2$ mm$^3$ (for $b$, $c$ and $a$ axes, respectively) are spot welded by 25 $\mu$m-diameter Pt wires to make the standard Hall-bar electric contacts on a single pulse Sunstone welder. For p-SnSe, the resulting ohmic contact-resistance is less than 10 $\Omega$. All the transport properties are measured in an Oxford-14 T cryostat, using Keithley 2400 and 2182 A as current sourcemeter and nanovoltagemeters, respectively. Seebeck coefficients are measured with a $T$ gradient of less than 0.5 K, determined by two E-type miniature thermocouples. Silver paint or epoxy causes Schottky contacts, and thus not suitable for low T measurements.

**DFT calculations**. The electronic structure of SnSe is calculated with the Vienna ab initio simulation package (VASP)[35,36] by the method of the projector augmented wave[37] and the generalised gradient approximation (GGA)[38], taking the exchange-correlation potential as well as spin-orbit coupling into account. The plane-wave cutoff energy is set at about 400 eV and the k-point sampling is performed by the

Monkhorst-Pack scheme[39]. The total energy is ensured to be converged within 0.002 eV per unit cell.

**Atomic force microscopy**. We use an ambient multi-mode Park NX10 AFM to characterise the surface morphology of freshly cleaved SnSe crystals. The point dislocation intensity is analysed by AFM imaging of 20 random locations spread over the whole crystal surface. For qPlus nc-AFM, we conduct in situ cleavage of bulk SnSe crystals followed by heating up the sample at 490 K for 2 h to create clean surface required for atomic-resolution imaging. The turning-fork based qPlus sensor was employed to perform the nc-AFM experiment at 4.3 K in ultrahigh vacuum conditions ($<2 \times 10^{-10}$ Torr). The qPlus nc-AFM images of SnSe crystal are measured in the constant amplitude mode, using frequency shift as the feedback set point ($\Delta f = -26$ Hz, resonance frequency $f_0 = 24.58$ kHz, resonance quality factor $Q = 12,000$ and constant amplitude $A = 0.5$ nm).

**Data availability**. The authors declare that the main data supporting the findings of this study are available within the paper and its Supplementary Information files. Extra data are available from the corresponding authors upon request.

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

## Acknowledgements
This work is supported by the National Key R&D Program of the MOST of China (Grant Nos. 2016YFA0300204, 2017YFA0303002 and 2016YFA0401000), and the National Science Foundation of China (Grant Nos. 11574264, 11574337 and 11227902). D.W.S. is supported by "Award for Outstanding Member in Youth Innovation Promotion Association CAS". Y.Z. acknowledges the start funding support from the Thousand Talents Plan.

## Author contributions
Z.S. and Z.W. synthesised and characterised the single crystals. Z.W., Z.S., F.S. and Q.H. performed the transport measurements. C.F., Z.S., Z.L., W.L, Y.H. and D.W.S. carried out the ARPES experiments. C.H. and Y.L. did the DFT calculations. Z.Q., Q.H., H.F. and J.L. conducted the AFM experiments. Z.W., C.F., D.W.S. and Y.Z. analysed the experimental data and wrote the paper with input from all authors. Y.Z. and D.W.S. supervised the project.

## Additional information

**Competing interests:** The authors declare no competing financial interests.

