## [Peer Review File · Nature Communications]

Reviewers' Comments:

Reviewer #1:

Remarks to the Author:

The authors report on various methods of growing single crystals of SnSe, on the controlled way of their doping and the obtained multi-valley nature of transport. The work is supported by DFT band structure calculations and the first ARPES measurements of the SnSe system. I particularly value the latter as it is essential to have experimental evidence of the form and shape of bands, which, so far, have been available only via theoretical estimates.

SnSe has gained a tremendous interest worldwide during the past couple of years for its record-high thermoelectric figure of merit achieved with single-crystalline forms of the material. Because single crystals are impractical for fabrication of thermoelectric modules operating as power generators, numerous attempts have been reported to replicate the performance of single-crystalline SnSe with polycrystalline forms of the structure. Unfortunately, the effort has not been successful yet. Part of the problem rests in an incomplete understanding of the exact form of the valence band structure where many proposals have been made but not verified by sharp spectroscopic probes, such as ARPES. The paper provides the first direct experimental evidence regarding the valence band structure. From that perspective, the paper should be welcomed and valued. The other topics covered in the paper, namely weak localization aspects of the low temperature transport and Shubnikov-de Haas oscillations, while interesting, are nothing new as they are often observed in reasonably pure semiconducting structures at low temperatures and used to characterize the Fermi surface. The authors often use non-standard phrases that may confuse the readers and make it difficult to understand what exactly they are trying to say regarding their results. I will go through the manuscript step-by-step and will point out the most glaring issues.

1. In the second sentence of Abstract they use: "... optimal electrical dosage..." . This should be replaced by "... optimal carrier concentration..." .
2. Here I am a bit picky but the statement in the Abstract: "...ranging from intrinsic semiconducting behavior to typical metal with carrier density of $1.23 \times 10^{18} \text{ cm}^{-3}$ at room temperature." is a bit misleading as metals do not have carrier densities as low as 10^{18} cm^{-3} but five orders of magnitude larger. Perhaps replacing the word "metal" by "degenerate semiconductor" would be appropriate.
3. The authors are grossly inconsistent in the designation of the crystallographic axes and use three different forms throughout the paper: they call the crystallographic axes a, b, c; they refer to them as "armchair" and "zig-zag"; and without any definition they write the Seebeck coefficient and the electrical conductivity with subscripts "xx", i.e., as S_{xx} and σ_{xx} . Please stick with one form of referring to the crystallographic axes and do not confuse the readers. I suggest using axes a, b, c, only. What does the "xx" mean? What direction is it? Moreover, in the definition of the dimensionless figure of merit ZT, they designate the thermal conductivity merely as κ , without any subscript as if it was arbitrary in what direction the thermal conductivity is measured. Remember, to get the correct figure of merit, all three transport parameters (S, σ , and κ) must be measured along the same crystallographic axis!
4. P.3, second last paragraph, the authors use "electrical doping". This should read either "electronic doping" or, if they wish to be specific to indicate that it is p-type doping, then use "hole doping" or "p-type doping".
5. In the same paragraph, the authors imply that the p-type character of SnSe and how it arises has not yet been considered. This is not so. The origin of the p-type nature of SnSe is a topic dealt with by G. Duvjir et al. in Applied Physics Letters 110, 262106 (2017). This paper should be referenced.
6. I would like to see in the main text the temperature dependence of the carrier concentration as obtained by Hall effect measurements. It is nice to show the mobility as a function of temperature, but this follows from combining the Hall effect data with the electrical conductivity data. In many respects, it is critical to understand the temperature dependence of the carrier concentration.
7. P.4, the authors state that the upturn in the electrical resistivity below 50 K as T decreases is a manifestation of the quantum interference effect, i.e., WL. Making such a definitive statement at

that stage of the paper, without yet presenting the magneto-transport data, is quite premature as the upturns on resistivity curves at low temperatures could be caused by a variety of phenomena, the most obvious being the carrier freeze out.

8. I am also puzzled why in Fig. 1f the mobility at the very lowest temperatures seems to increase as the temperature decreases rather than keep decreasing.

9. I do not think the term “pudding –mold” model is sufficiently clear and universally used and the authors should at least sketch what they mean by it or refer to a shape of the curve in Fig. 2h.

10. On p.5, the authors state: “By utilizing different growth methods and fine tuning growth parameters, we demonstrate that the extrinsic doping in SnSe can be widely tuned from semiconducting to $1.23 \times 10^{18} \text{ cm}^{-3}$.” First of all, what do they mean by “extrinsic doping”? Do they mean that the carrier concentration is adjusted by stoichiometry rather than by actual doping using foreign species? If so, it is improper to use the word extrinsic. Furthermore, the carrier concentration of $1.23 \times 10^{18} \text{ cm}^{-3}$ is a perfectly typical carrier concentration in a degenerate semiconductor and the sentence then makes little sense.

11. The presence of the SnSe₂ phase is the key to the authors’ argument why SnSe is a p-type material, yet they provide no evidence of its existence in the main body of the paper. By the way, if the micro-domains of SnSe₂ were numerous, there must be regions where the structure is poor in Se, given the initial stoichiometric quantities of Sn and Se. What is then the effect of such Se vacancies?

12. On p.15, the authors state: “...it is also important to take into account the unique point defect dislocation defects, which have drastically changed the 2D nature of ideal SnSe as evident by the quantum oscillations results. In the vicinity of these point defects, the local Sn-Se bonding network are likely to be disturbed by the reversing of stacking sequences between neighboring MLs, leading to changes in the hybridization of the Se 4p- and Sn 5s-orbitals.” Where is any evidence for such stacking sequence reversals? By the way, how do dislocation defects affect the quantum oscillations? How is it revealed? While dislocations have aspects of point defects, they also create extended strain fields in their proximity. What is the effect of such dislocation strain fields on the transport?

13. There are numerous typos and incorrect phrases that should be taken care of.

14. The legend of some of the figures (e.g., Fig. 2e-2f) is too small to read comfortably and the font should be increased.

Overall, at this stage, I am not convinced that the paper is suitable for a high impact journal, such as Nature of Communications.

Reviewer #2:

Remarks to the Author:

Recent establishment of the extremely high ZT in SnSe (Nature 2014, 508, 373 and Science 2016, 351, 141), a simple, non-toxic, single crystalline material, has revolutionized and triggered enormous enthusiasm in the field of thermoelectrics. Such extraordinary thermoelectric performance over a broad temperature range is undoubtedly related to its underlying electronic structure.

The manuscript NCOMMS-17-17020, titled “Defects controlled hole doping and multi-valley transport in SnSe single crystals”, by Zhen Wang et al. provides a unique perspective in understanding the fascinating thermoelectric performance of SnSe, by revealing the critical importance of defects in determining the electronic structures and electrical properties of SnSe single crystals. They have identified two major types of defects, namely point dislocations and SnSe₂ micro-domains induced by local phase segregations, by complementary experimental techniques. Based on statistics on various batches of samples prepared by varying growth methods and parameters, they have shown that SnSe₂ micro-domains are responsible for self hole-doping in SnSe, while point dislocations interconnect second-nearest neighboring monolayers and creating 3D-like Fermi surfaces. Such claims are well supported by angle-resolved photoemission spectroscopy, which would be the first experimental observation of multi-valley

valence band in SnSe (named as “pudding-mold” band in the paper; See also arXiv: 1707.04289 which appears later), and by quantum transport, which reveals strong weak localization and surprising interlayer coupling strength.

The manuscript has the potential to be a highly cited and impactful work in the emergent field of monochalcogenide-based thermoelectric materials. The results presented in this work are very encouraging by suggesting that the ZT value of SnSe may still have potential to be further enhanced by either introducing more hole carriers, or by defect engineering.

However, before the acceptance of the manuscript, we would like to suggest the authors to make the following changes to clarify several important issues, which we believe that they should be answered for the benefits of the general audience of Nature Communications:

1. For the existence of multi-valley band, or “pudding-mold” band, it is expected to see some substantial changes in the dependence of Seebeck (S_{xx}) coefficients on doping levels (n), which would be $S_{xx} \sim n^{-2/3}$ for single parabolic band in conventional doped semiconductors. Since the authors have different batches of crystals with varying doping concentrations, we would like to ask them provide such S_{xx} and n scaling relation.

2. Page 4, Figure 1d, is the energy level alignment diagram based on theoretical calculations or it is deduced from experimental results. The authors need to clarify this point.

3. Page 6, Paragraph 2, Lines 11-12, “...SnSe₂ micro crystals, which are typically several micrometers in lateral sizes and several MLs to several tens MLs in thickness (Figure S6 in SI)”, the authors should provide a statistics on the thickness of SnSe₂ micro-domains, which in principle can be done by AFM.

4. Is the claim of “SnSe₂ micro-domains” based on experimental results? Can it be non-stoichiometric SnSe_{2-x}? We have noticed that the authors have done Raman analysis on SnSe crystal, and on exfoliated SnSe₂ and SnSe flakes. Can the authors carry out Raman experiments directly on SnSe₂/SnSe heterostructures (See samples in Figure S6).

5. Page 14, Paragraph 2, Lines 16-18, “For SF1 SnSe, the dislocation density is larger than $4.5 \times 10^4 \text{ mm}^{-2}$, which corresponds to more than one dislocation site in a randomly searched area of $10 \times 10 \mu\text{m}^2$ (see SI for AFM statistics).”, the AFM statistics of point dislocations is not found in the supplementary information. Please add this Supplementary Figure in the revision.

6. Figure 4d, adding some labels (1, 2, 3 and 4 like the inset) to indicate the stacking order of different monolayers will help the readers to understand the formation of point dislocation.

Reviewer #3:

Remarks to the Author:

The manuscript presents highly anisotropic electronic structure of SnSe investigated by using angle-resolved photoemission spectroscopy and quantum transport, where a unique “pudding-mold” shaped valence band with quasi-linear energy dispersion is revealed. The authors propose that the self-hole doping mechanism in SnSe is controlled by the formation of SnSe₂ micro-domains induced by local phase segregation, which effectively introduces hole carriers in the SnSe bulk by interfacial charge transferring. The presentation is quite clear and the explanation to physical mechanisms is novel and reasonable. In particular, this study would further promote studies on electronic structure, transport properties and doping mechanism into p-doping SnSe. Therefore, I recommend the publication when revised.

1. The formation and characteristics of the point dislocation are not clearly described in this paper. It is better to give a detailed description of the geometric features and the local phase segregation of SnSe₂.

2. The authors have investigated the band structure of p-SnSe measured by ARPES and calculated with DFT theory. Although most of the DFT and ARPES results are well consistent, the detailed band structure in the vicinity of the EF of SnSe still shows some pronounced differences. The authors may consider using the tight-binding model to further confirm their results.

3. To provide more information for readers, the following paper can be cited:
Thermoelectric properties of single-layered SnSe sheet
Nanoscale, 2015, 7, 15962

I. Replies to the Referees' comments

Reviewer #1:

Reviewers' comments:

Reviewer #1 (Remarks to the Author):

Comment 1-1: The authors report on various methods of growing single crystals of SnSe, on the controlled way of their doping and the obtained multi-valley nature of transport. The work is supported by DFT band structure calculations and the first ARPES measurements of the SnSe system. I particularly value the latter as it is essential to have experimental evidence of the form and shape of bands, which, so far, have been available only via theoretical estimates.

SnSe has gained a tremendous interest worldwide during the past couple of years for its record-high thermoelectric figure of merit achieved with single-crystalline forms of the material. Because single crystals are impractical for fabrication of thermoelectric modules operating as power generators, numerous attempts have been reported to replicate the performance of single-crystalline SnSe with polycrystalline forms of the structure. Unfortunately, the effort has not been successful yet. Part of the problem rests in an incomplete understanding of the exact form of the valence band structure where many proposals have been made but not verified by sharp spectroscopic probes, such as ARPES. The paper provides the first direct experimental evidence regarding the valence band structure. From that perspective, the paper should be welcomed and valued.

Reply 1-1: We strongly agree with the Referee that a comprehensive understanding of the electronic structure of SnSe is the key to decipher and improve the thermoelectric performance of SnSe. Very surprisingly, despite that the record-breaking ZT value of SnSe is published in as early as 2014, so far there has no experimental report of the energy band structure of SnSe using ARPES, which is one of the most straightforward spectroscopic methods in determining solids' electronic structure.

For the first time, our work determined the main pudding-mold shaped valence band structure of p-type SnSe single crystals, most noticeably showing multiple valleys with quasi-linear energy dispersion near the Fermi level. Such finding provides insights into the extraordinary thermoelectric properties of SnSe. More critically, using complementary techniques of atomic force microscope and quantum transport, we have revealed that such p-type doping and 3D Fermi surface in as-synthesised SnSe are controlled by two peculiar types of defects. We believe that these findings will greatly advance the field, and may pave a new way in controlling electronic doping and improving thermoelectric performance in SnSe by defect engineering.

We have noticed that just following our work (arXiv:1706.10054), Pletikoscic in collaboration with Prof. R. J. Cava from Princeton Univ. soon posted a manuscript of ARPES study on SnSe (arXiv:1707.04289). Although these results are in general consistent with our work, Pletikoscic *et al.* did not resolve the full Brillouin zone of SnSe.

Comment 1-2: The other topics covered in the paper, namely weak localization aspects of the low temperature transport and Shubnikov-de Haas oscillations, while interesting, are nothing new as they are often observed in reasonably pure semiconducting structures at low temperatures and used to characterize the Fermi surface.

Reply 1-2: Regarding *Referee A's* concern on the significance of the charge transport results of our work, we should point out that the key findings of this work are two folded: the electronic structure basis of the unprecedented thermoelectric properties in p-SnSe; and more critically, the unique defect physics in determining the electronic structures and electrical properties of p-SnSe.

First of all, the ARPES revealed quasi-linear energy band dispersion and multi-valley Fermi surface in p-type SnSe are indeed one of the key discoveries of our work, and provide the basis in understanding the thermoelectric performance of this material. However, the **well-known surface sensitivity drawback** of ARPES naturally leads to the uncertainty whether the determined electronic structure is rooted in the bulk, or it is intimately related to surface states. The **complementary** DFT calculations and angle-dependent Shubnikov-de Haas (SdH) oscillation results successfully reproduced the multi-valley Fermi surface and effective mass determined by ARPES, and thus allow us to consequently finalizes our model.

Charge transport study is also indispensable in the finding of defect controlled hole doping in p-SnSe, whose effects on thermoelectric performance have been effectively analyzed by weak localisation induced negative magnetoresistance at low temperature. The SdH oscillations as well reveal an unexpected 3D Fermi surface morphology, which is in stark contrast to the theoretical predictions of very weak interlayer coupling. These quantum transport results are critical to pin down the defect origins of the electronic structure and electrical properties of p-SnSe.

Last but not least, with the unique multi-valley Fermi surface along the *c*-axis (armchair), the experimental phenomena of quantum transport in p-SnSe single crystals are not mundane at all. These findings have effectively deepened our understanding of the novel properties of this material and might pave the way for further improvement of its thermoelectric performance. Some examples are listed below:

- I. Non-saturating negative magnetoresistance (NMR) for $I // c$ and $H // a$, which has never been reported before. The high anisotropic behavior in NMR are in coincidence with the geometry of the 'pudding-mold' pockets, and thus is closely correlated to the multi-valley transport in p-SnSe.
- II. Our preliminary results show unconventional T^2 dependence of Dingle temperature, which may be a manifestation of the unharmonic phonon modes in SnSe. Giant phonon unharmonicity has been regarded as an important fact in determining the ultrahigh ZT value. Direct probing of the interaction between charge carriers and phonon modes may greatly advance our understanding of ZT in resonant p-bonding systems.
- III. Due to very small charge carrier pockets, with a field of about 9 T, the SdH oscillations in p-SnSe have reached the quantum limit (QL). Although not directly correlated to thermoelectric, it would be interesting to try high field to study various exotic physical phenomena such as anomalous MR and Hall, spontaneous symmetry breaking, and quantum linear magnetoresistance etc.

Comment 1-3: The authors often use non-standard phrases that may confuse the readers and make it difficult to understand what exactly they are trying to say regarding their results. I will go through the manuscript step-by-step and will point out the most glaring issues.

Reply 1-3: We gratefully thank the Referee for the expert reviewing of our manuscript. We have addressed the Referee's concerns by revising the paper accordingly, fully taking his/her comments into considerations. The changes are summarized below in a point-to-point fashion.

Comment 1-3-1: In the second sentence of Abstract they use: "...optimal electrical dosage..." . This should be replaced by "...optimal carrier concentration..." .

Reply 1-3-1: We have followed this suggestion by deleting "optimal electrical dosage" in our revised manuscript.

See: Page 2, Abstract.

Comment 1-3-2: Here I am a bit picky but the statement in the Abstract: "...ranging from intrinsic semiconducting behavior to typical metal with carrier density of $1.23 \times 10^{18} \text{ cm}^{-3}$ at room temperature." is a bit misleading as metals do not have carrier densities as low as 10^{18} cm^{-3} but five orders of magnitude larger. Perhaps replacing the word "metal" by "degenerate semiconductor" would be appropriate.

Reply 1-3-2: We agree with the Referee, and we have replaced the word "metal" in the main text by "degenerate semiconductor" following his/her advice. Note that the original sentence has been moved to the introduction, since the hole doping level ($1.23 \times 10^{18} \text{ cm}^{-3}$) is not the key finding of this study.

See:

- I. Page 4, Paragraph 2, Line 11-13, "By utilising different growth methods and fine tuning parameters, we demonstrate that the extrinsic hole doping in SnSe can be widely tuned from semiconducting to $1.23 \times 10^{18} \text{ cm}^{-3}$."
- II. Page 5, Paragraph 2, Line 2-3, "..., ranging from semiconductivity to p-type degenerate semiconductors."

Comment 1-3-3: The authors are grossly inconsistent in the designation of the crystallographic axes and use three different forms throughout the paper: they call the crystallographic axes a, b, c; they refer to them as "armchair" and "zig-zag" and without any definition they write the Seebeck coefficient and the electrical conductivity with subscripts "xx", i.e., as S_{xx} and σ_{xx} . Please stick with one form of referring to the crystallographic axes and do not confuse the readers. I suggest using axes a, b, c, only. What does the "xx" mean? What direction is it? Moreover, in the definition of the dimensionless figure of merit ZT, they designate the thermal conductivity merely as κ , without any subscript as if it was arbitrary in what direction the thermal conductivity is measured. Remember, to get the correct figure of merit, all three transport parameters (S, σ , and κ) must be measured along the same crystallographic axis!

Reply 1-3-3: We appreciate the Referee's good suggestion. In general, we agree with him that the

original symbols and subscript labeling are not clear enough. As requested by the Referee, we have adopted **a-, b-, c-axes** to represent crystallographic directions, and we have also used **a, b, c** as subscript labeling when referring to crystallographic dependent physical quantities, such as S_a , S_b , σ_{ab} (diagonal axis of the **a-b** plane), etc. Please note that it is conventional to use S_{xx} to represent Seebeck coefficient, while S_{xy} stands for Nernst coefficient. Such a naming convention is an analogy of longitudinal conductivity (σ_{xx}) and transverse Hall conductivity (σ_{xy}). For highly anisotropic SnSe-type compounds, such naming rules are indeed not appropriate. The detailed changes to the manuscript are summarized below:

- I. Page 3, Paragraph 1, Line 5-6, Line 15, Line 17, and Line 20, we have removed all xx -subscript labeling in S , σ , κ .
- II. Page 7, Paragraph 2, Line 4, Line 6, Line 8, S_{xx} to S_b .
- III. Page 8, Paragraph 2, Line 3, Line 18, S_{xx} to S .
- IV. Page 12, Paragraph 1, Line 7, Line 11, S_{xx} to S .
- V. Page 20, Figure 1, Caption (e), S_{xx} to S_b .
- VI. Page 21, Figure 2, Caption (h), S_{xx} to S .
- VII. We have added sentences to highlight the crystallographic anisotropy in SnSe, and the critical importance of systematic charge transport study.

See: Page 3, Paragraph 2, Line 4-9, “Remarkably, the resonant p-bonding network in the SnSe family forms unique puckering structure similar to black phosphorus (BP) [14], which is known to show distinct anisotropic physical properties along different crystallographic axes [15,16]. In SnSe, highly anisotropic thermoelectric properties, i.e. S , σ and κ , have also been observed with a strong doping dependence [2,6]. However, a systematic charge transport study in complementary to the electronic structure has not been achieved yet.”

Comment 1-3-4: P.3, second last paragraph, the authors use “electrical doping”. This should read either “electronic doping” or, if they wish to be specific to indicate that it is p-type doping, then use “hole doping” or “p-type doping” .

Reply 1-3-4: We thank the Referee for this useful comment, and we have changed the phrase to “**hole doping**”.

See: Page 3, Paragraph 2, Line 1.

Comment 1-3-5: In the same paragraph, the authors imply that the p-type character of SnSe and how it arises has not yet been considered. This is not so. The origin of the p-type nature of SnSe is a topic dealt with by G. Duvjir et al. in Applied Physics Letters 110, 262106 (2017). This paper should be referenced.

Reply 1-3-5: We thank the Referee for reminding us this excellent work. During the preparation of our manuscript, we were not aware of this paper since it had not been formally published. As suggested by the Referee, we have added the aforementioned APL paper as **Reference 17**, which has been cited accordingly in the main text. Related changes are the following:

- I. Page 16, the “Reference Section”, new **Ref. [17]**, Duvjir G. *et al.* Origin of p-type characteristics in a SnSe single crystal. *Appl. Phys. Lett.* **110**, 262106 (2017).

II. Page 4, Paragraph 1, Line 1-2, we added the sentence “Using scanning tunneling microscopy, Duvjir *et al.* suggest that Sn vacancies may play a decisive role in self hole doping [17].”

However, we must emphasize here that the Reference work mainly focuses on the effect of Sn/Se vacancies on the p-type doping in SnSe. This experimental discovery is important but **should not** be attributed to be the main self-hole doping mechanism in SnSe. Indeed, we barely discover vacancies in our stoichiometric single crystals with high quality, as evident by the q-plus based nc-AFM results. It is also noticeable that for Sn-rich crystals (batch SF12), which may have high density of Se vacancies, the measured samples are still p-doped, but not n-type as suggested by Ref. [17]. Instead of vacancies, the formation of SnSe₂ micro-domains would hold the key to understand hole doping in SnSe, which is one of the key findings of our work.

Comment 1-3-6: I would like to see in the main text the temperature dependence of the carrier concentration as obtained by Hall effect measurements. It is nice to show the mobility as a function of temperature, but this follows from combining the Hall effect data with the electrical conductivity data. In many respects, it is critical to understand the temperature dependence of the carrier concentration.

Reply 1-3-6: As suggested by the Referee, we have added the T -dependent Hall coefficient in the main text as the inset of Figure 1f.

Comment 1-3-7: P.4, the authors state that the upturn in the electrical resistivity below 50 K as T decreases is a manifestation of the quantum interference effect, i.e., WL. Making such a definitive statement at that stage of the paper, without yet presenting the magneto-transport data, is quite premature as the upturns on resistivity curves at low temperatures could be caused by a variety of phenomena, the most obvious being the carrier freeze out.

Reply 1-3-7: The correlation between the resistivity upturn below 50 K and weak localisation is unambiguously supported by the magneto-transport data, most noticeably, T -dependent negative magnetoresistance (NMR) at low fields. Detailed analyses of NMR can be found in the Supplemental Information (See Supplementary Note 9 for detailed weak localization analysis in SI).

As requested by the Referee, we have moved the sentence to the subsection of “**Quantum localisation in p-SnSe**”. See Page 9, Paragraph 1, “The resistivity upturn of p-SnSe below 50 K is a transport signature of weak localization (WL), which is evident by NMR in low fields when constructive quantum interference is suppressed by breaking time reversal symmetry [25].”

Note that the “carrier freeze out” process is only valid for thermally activated charge carriers in semiconductors, which is the case for batch SF10, SF8, SF5 and PVD. For the metallic batches (see Table I in the manuscript), such hypothesis can be easily rejected by Hall measurements, which only show few percentage changes in carrier density from 300 K to 1.5 K, typical behavior for semimetals/metals. The T -dependent Seebeck coefficients of our p-SnSe crystals also show typical metallic behavior, inconsistent with the “carrier freeze out” suggestion.

Comment 1-3-8: I am also puzzled why in Fig. 1f the mobility at the very lowest temperatures seems to increase as the temperature decreases rather than keep decreasing.

Reply 1-3-8: From the T -dependent quantum oscillation measurement (see Supplementary Fig. 10 in SI), we find that the SdH oscillation frequency shifts significantly as T decreases below 20K, indicating that the Fermi level (E_F) of the system is moving toward higher energy. This is consistent with the T -dependent carrier concentration measurements, as shown in the inset of Figure 1f as well as in Supplementary Fig. 8 and Fig. 9. Such downward shifting in E_F will lead to smaller effective mass, when the energy band becomes quasi-linear away from the VB maximum. The E_F shifting provides a reasonable explanation on the observed mobility upturn below 20 K, and the phenomena are worthy of further experimental investigation by introducing an external gate to tune the E_F energy continuously.

Comment 1-3-9: I do not think the term “pudding-mold” model is sufficiently clear and universally used and the authors should at least sketch what they mean by it or refer to a shape of the curve in Fig. 2h.

Reply 1-3-9: We use the phrase “*pudding-mold*” band to refer the peculiar shape of the valence band of SnSe, which has two equivalent VB maximums in close proximity along the high-symmetry Γ -Z line and quasi-linear energy dispersion away from the relative flat band tops. The term was first introduced by Kuroki and Arita to explain the large thermoelectric performance in Na_xCoO_2 (Ref. [24]), which has now been widely accepted as one of the main mechanisms for high Seebeck coefficients. This is evident by the high citations of this work (99 times since it was formally published in 2007) for the explanation of giant Seebeck coefficients in FeAs_2 [PRB, 88(7), 075140 (2013)], LiRh_2O_4 [PRB, 78(11), 115121 (2008); PRL 101, 086404(2008)], PtSb_2 [APL 100, 252104 (2012)], and a variety of other compounds.

While, we agree with the Referee that a clear description of the term would further improve the readability of the manuscript. As a response to the Referee’s suggestion, we have added several sentences to the related parts.

See, Page 4, Paragraph 2, Line 5-9, “Using high-resolution ARPES, we have resolved for the first time the unique *pudding-mold* shaped valence band (VB) of p-SnSe, characterised by two equivalent VB maximums in close proximity along the high-symmetry Γ -Z line and quasi-linear energy dispersion away from the relatively flat band tops.”

Comment 1-3-10: On p.5, the authors state: “By utilizing different growth methods and fine tuning growth parameters, we demonstrate that the extrinsic doping in SnSe can be widely tuned from semiconducting to $1.23 \times 10^{18} \text{ cm}^{-3}$.” First of all, what do they mean by “extrinsic doping”? Do they mean that the carrier concentration is adjusted by stoichiometry rather than by actual doping using foreign species? If so, it is improper to use the word extrinsic. Furthermore, the carrier concentration of $1.23 \times 10^{18} \text{ cm}^{-3}$ is a perfectly typical carrier concentration in a degenerate semiconductor and the sentence then makes little sense.

Reply 1-3-10: Here, we use the word “**extrinsic**” to emphasize that the hole charge carriers are introduced by the formation of SnSe₂ micro-domains in the crystals. By minimizing or eliminating the local phase segregation process, which leads to the formation of SnSe₂ micro-domains, we have synthesized “**intrinsic**” semiconducting SnSe, as summarized in Table I. In another word, we want to highlight the fact that SnSe is not intrinsically p-type semiconductor, as claimed by many previous literatures. Accordingly, we have changed the phrase “**extrinsic doping**” to “**hole doping**”.

For the second question, we agree with referee that it would be more appropriate to be named as degenerate semiconductor. See **Reply 1-3-2**.

Comment 1-3-11: The presence of the SnSe₂ phase is the key to the authors’ argument why SnSe is a p-type material, yet they provide no evidence of its existence in the main body of the paper. By the way, if the micro-domains of SnSe₂ were numerous, there must be regions where the structure is poor in Se, given the initial stoichiometric quantities of Sn and Se. What is then the effect of such Se vacancies?

Reply 1-3-11: The existence of SnSe₂ in p-SnSe single crystals is unambiguously proved by complementary experimental results, such as optical microscopy, ambient AFM, SKPM and Raman. (See Supplementary Fig. 4, Fig. 6 and Fig. 7 in SI).

In response to the Referee’s advice, we have added a typical optical photo of SnSe₂/SnSe heterostructure in the inset of Figure 1d.

Regarding the second question, the local phase segregation induced SnSe₂ micro-domain formation is not necessarily to cause Se vacancies in SnSe single crystals. Such a claim is solidly based on q-plus nc-AFM imaging of the SF1-SnSe samples, which very rarely show vacancies with random scanning locations, as shown in Figure 1b. This may be due to the 2D nature of SnSe and SnSe₂, which intends to accumulate excessive Sn at interlayer positions. We have added an AFM image showing Sn rich surface in the vicinity of SnSe₂ micro-domains. See Supplementary Fig. 19 and related discussions in the SI.

Nevertheless, it would be interesting to control Sn/Se vacancy formation in SnSe. For example, in the new Ref. [17] [APL 110, 262106 (2017)], STM results and DFT calculations suggest that Se vacancies can create localized electron near the vacancy sites, while Sn vacancies are hole donors. On the contrary, we have observed strong hole doping in Se-deficient SnSe crystals (Batch SF12), which show negligible SnSe₂ micro-domains in the bulk. However, it will require a systematic study on stoichiometry, growth methods and detailed parameter tuning to get insights into the Sn/Se vacancy physics, which could be potentially explored in another follow-up project.

Comment 1-3-12: On p.15, the authors state: “...it is also important to take into account the unique point defect dislocation defects, which have drastically changed the 2D nature of ideal SnSe as evident by the quantum oscillations results. In the vicinity of these point defects, the local Sn-Se bonding network are likely to be disturbed by the reversing of stacking sequences between neighboring MLs, leading to changes in the hybridization of the Se 4p- and Sn 5s-orbitals.” Where is any evidence for such stacking sequence reversals? By the way, how do dislocation defects affect the quantum oscillations? How is it revealed? While dislocations have aspects of

point defects, they also create extended strain fields in their proximity. What is the effect of such dislocation strain fields on the transport?

Reply 1-3-12: We have updated the Figure 4d to better illustrate the reversing of stacking sequence (not stacking order!) induced by a point dislocation. As shown in the inset of Figure 4d, on the left side of the point dislocation, three SnSe ML are stacked in the antiferroelectric order with the labeling of ML1, ML2 and ML3, respectively, from the top to the bottom. With the formation of a dislocation point, ML1 and ML3 is interconnected in the vicinity of the point defect, leading to the reversing of stacking sequence. The formation of point dislocations in SnSe is required by the antiferroelectric ordering of the bulk, which prohibits two MLs with the same ferroelectric dipole orientation to be stacked face-to-face directly (see the Figure attached below). High density of point dislocations interweave SnSe along the a -axis, greatly weakening the two-dimensionality, as manifested by the 3D Fermi surface in quantum oscillations. This has been detailed discussed in the manuscript.

Regarding the understanding of local strain field created by point dislocations, it will require expertized nanomechanical simulations, which would be a fascinating follow-up project to be explored [examples of nanomechanical modelling of 2D materials, see PRL 106, 255503 (2011); Nat. Phys. 8, 739 (2012), and PRL 114, 065501 (2015)]. For the reference of potential readers, these references have been added into the main text. See: Page 17, “Reference Section”, Ref. [29]-[31].

Comment 1-3-13: There are numerous typos and incorrect phrases that should be taken care of.

Reply 1-3-13: We thank the referee and we have carefully proofread the manuscript and corrected

the typos.

Comment 1-3-14: The legend of some of the figures (e.g., Fig. 2e-2f) is too small to read comfortably and the font should be increased.

Reply 1-3-14: We have updated the legends in Figure 2 with larger symbols and labels.

Again, we would like to thank Referee A for his/her thoughtful comments and suggestions, which we think have helped to greatly improve the readability and clarity of our manuscript.

Reviewer #2 (Remarks to the Author):

Recent establishment of the extremely high ZT in SnSe (Nature 2014, 508, 373 and Science 2016, 351, 141), a simple, non-toxic, single crystalline material, has revolutionized and triggered enormous enthusiasm in the field of thermoelectrics. Such extraordinary thermoelectric performance over a broad temperature range is undoubtedly related to its underlying electronic structure.

The manuscript NCOMMS-17-17020, titled “ Defects controlled hole doping and multi-valley transport in SnSe single crystals” , by Zhen Wang et al. provides a unique perspective in understanding the fascinating thermoelectric performance of SnSe, by revealing the critical importance of defects in determining the electronic structures and electrical properties of SnSe single crystals. They have identified two major types of defects, namely point dislocations and SnSe₂ micro-domains induced by local phase segregations, by complementary experimental techniques. Based on statistics on various batches of samples prepared by varying growth methods and parameters, they have shown that SnSe₂ micro-domains are responsible for self-hole-doping in SnSe, while point dislocations interconnect second-nearest neighboring monolayers and creating 3D-like Fermi surfaces. Such claims are well supported by angle-resolved photoemission spectroscopy, which would be the first experimental observation of multi-valley valence band in SnSe (named as “pudding-mold” band in the paper; See also arXiv: 1707.04289 which appears later), and by quantum transport, which reveals strong weak localization and surprising interlayer coupling strength.

The manuscript has the potential to be a highly cited and impactful work in the emergent field of monochalcogenide-based thermoelectric materials. The results presented in this work are very encouraging by suggesting that the ZT value of SnSe may still have potential to be further enhanced by either introducing more hole carriers, or by defect engineering.

However, before the acceptance of the manuscript, we would like to suggest the authors to make the following changes to clarify several important issues, which we believe that they should be answered for the benefits of the general audience of Nature Communications:

We appreciate the critical and very encouraging comments from the Referee, and we would like to thank the referee for his/her expertized reviewing of our manuscript.

Comment 2-1: For the existence of multi-valley band, or “pudding-mold” band, it is expected to see some substantial changes in the dependence of Seebeck (S_{xx}) coefficients on doping levels (n), which would be $S_{xx} \sim n^{-2/3}$ for single parabolic band in conventional doped semiconductors. Since the authors have different batches of crystals with varying doping concentrations, we would like to ask them provide such S_{xx} and n scaling relation.

Reply 2-1: Within the limited time, we have measured as-many-as-possible samples to get Seebeck coefficients for different batches of SnSe single crystals at room temperature. The results are plotted as a function of carrier concentration as shown in the Figure below (added as Supplementary Fig. 12 in SI). It is clear that Seebeck coefficients of p-SnSe does not follow the $-2/3$ power law scaling as a function of doping, which is not surprising since the $n^{-2/3}$ behavior is for **single parabolic band and energy-independent scattering** system. In the following, we will study T -dependent S vs n relations, which may allow us to get insights into the unharmonic phonon modes.

Comment 2-2: Page 4, Figure 1d, is the energy level alignment diagram based on theoretical calculations or it is deduced from experimental results. The authors need to clarify this point.

Reply 2-2: The energy level alignment diagram is based on scanning Kelvin probe microscopy, which shows excellent agreement with Density functional theory calculations. We have added one new figure in SI (See Supplementary Fig. 7), showing SKPM determined work functions of bare SnSe surface and few-layer SnSe₂/SnSe heterostructure on the same sample.

Comment 2-3: Page 6, Paragraph 2, Lines 11-12, “...SnSe₂ micro crystals, which are typically several micrometers in lateral sizes and several MLs to several tens MLs in thickness (Figure S6 in SI)”, the authors should provide a statistics on the thickness of SnSe₂ micro-domains, which in principle can be done by AFM.

Reply 2-3: The statistics on the lateral sizes of SnSe₂ micro-domains are done using optical microscope. For the thickness analysis, we mainly use optical contrast calculations (which provide

high accuracy and efficiency). The optical-contrast based thickness is further confirmed by AFM imaging and profiling (See Supplementary Fig. 18 and Supplementary Note 10 in SI).

Comment 2-3: Is the claim of “SnSe₂ micro-domains” based on experimental results? Can it be non-stoichiometric SnSe_{2-x}? We have noticed that the authors have done Raman analysis on SnSe crystal, and on exfoliated SnSe₂ and SnSe flakes. Can the authors carry out Raman experiments directly on SnSe₂/SnSe heterostructures (See samples in Figure S6).

Reply 2-3: The claim of SnSe₂ micro-domains is unambiguously based on energy dispersive X-ray spectroscopy (EDS) and Raman measurements. Element analysis using EDS shows that the ratio of Sn:Se for these micro-domains is very close to 1:2, indicating its 1T-MoS₂ type crystal structure. As suggested by the referee, we have also measured the Raman spectroscopy of few-layer SnSe₂/SnSe heterostructures directly as shown in figure below. The Raman results of the heterostructures show fingerprinting peaks of SnSe₂ at 110 cm⁻¹ and 189 cm⁻¹, consistent with the EDS data. Note that the SnSe₂ peaks are superimposed on the strong background signals of bulk SnSe crystal.

Comment 2-4: Page 14, Paragraph 2, Lines 16-18, “For SF1 SnSe, the dislocation density is larger than $4.5 \times 10^4 \text{ mm}^{-2}$, which corresponds to more than one dislocation site in a randomly searched area of $10 \times 10 \text{ } \mu\text{m}^2$ (see SI for AFM statistics).”, the AFM statistics of point dislocations is not found in the supplementary information. Please add this Supplementary Figure in the revision.

Reply 2-4: We thank the Referee for his careful reading of our manuscript. The aforementioned Supplementary Figure has been added into the SI (See Supplementary Fig. 18).

Comment 2-5: Figure 4d, adding some labels (1, 2, 3 and 4 like the inset) to indicate the stacking order of different monolayers will help the readers to understand the formation of point dislocation.

Reply 2-5: We appreciate the Referee advice. We have updated the Figure 4d to better illustrate the reversing of stacking sequence induced by a point dislocation, with proper numeric labels indicating stacking sequence. See also **Reply 1-3-14**.

Reviewer #3 (Remarks to the Author):

The manuscript presents highly anisotropic electronic structure of SnSe investigated by using angle-resolved photoemission spectroscopy and quantum transport, where a unique “pudding-mold” shaped valence band with quasi-linear energy dispersion is revealed. The authors propose that the self-hole doping mechanism in SnSe is controlled by the formation of SnSe₂ micro-domains induced by local phase segregation, which effectively introduces hole carriers in the SnSe bulk by interfacial charge transferring. The presentation is quite clear and the explanation to physical mechanisms is novel and reasonable. In particular, this study would further promote studies on electronic structure, transport properties and doping mechanism into p-doping SnSe. Therefore, I recommend the publication when revised.

We appreciate the insightful comments from the Referee.

Comment 3-1: The formation and characteristics of the point dislocation are not clearly described in this paper. It is better to give a detailed description of the geometric features and the local phase segregation of SnSe₂.

Reply 3-1: In the revised manuscript, we have discussed the formation of the point dislocations. We have updated the Figure 4d to better illustrate the reversing of stacking sequence induced by a point dislocation, with proper labels indicating stacking sequence. As shown in the inset of Figure 4d, on the left side of the point dislocation, three SnSe ML are stacked in the antiferroelectric order with the labeling of “ML1”, “ML2” and “ML3”, respectively, from the top to the bottom. With the formation of a dislocation point, ML1 and ML3 is interconnected in the vicinity of the point defect, leading to the reversing of stacking sequence. The formation of point dislocations in SnSe is required by the antiferroelectric ordering of the bulk, which prohibits two MLs with the same ferroelectric dipole orientation to be stacked face-to-face directly. High density of point dislocations interweave SnSe along the *a*-axis, greatly weakening the two-dimensionality, as manifested by the 3D Fermi surface in quantum oscillations. This has already been detailed discussed in the manuscript.

See, Page 11, Paragraph 2, Line 16-27, and Supplementary Fig. 18 and Note 10 in SI.

Comment 3-2: The authors have investigated the band structure of p-SnSe measured by ARPES and calculated with DFT theory. Although most of the DFT and ARPES results are well consistent, the detailed band structure in the vicinity of the EF of SnSe still shows some pronounced differences. The authors may consider using the tight-binding model to further confirm their results.

Reply 3-2: We thank the Referee for the good suggestion. In response, we have added a new Supplementary Fig. 20 to highlight the pronounced differences in valence band structure between ARPES and DFT. The DFT results are further compared with the calculations using the tight-binding calculations (see Supplementary Fig. 20c, 20d and 20e), which essentially reproduce

the DFT results.

Comment 3-3: To provide more information for readers, the following paper can be cited:

Thermoelectric properties of single-layered SnSe sheet
Nanoscale, 2015, 7, 15962

Reply 3-3: We thank the referee for reminding us this excellent theoretical calculation work on single-layer SnSe. Micro-mechanical exfoliation and few-layer thermoelectric devices are indeed very important direction for the SnSe family. In the 2D limit, the binary BP-type lattice may offer rich physics to be discovered. In the revised manuscript, we have added the aforementioned paper as a reference (see Page 16, “Reference” section, Ref. [19]).

List of main changes to the manuscript which are not covered by the point-by-point replies to the Referees:

1. We have added three co-authors, namely, Zhengtai Liu, Wanling Liu and Yaobo Huang, who have made significant contributions to the ARPES experiments.
2. All references have been updated with the Nature-style reference format.
3. Format changes following the “Manuscript Checklist”.
4. Reference 13, the Preprint arXiv no. has been updated with the formal publication information. See Page 16, Ref. [13] “Mori, H., Usui, H., Ochi, M. & Kuroki, K. Temperature- and doping-dependent roles of valleys in the thermoelectric performance of SnSe: A first-principles study. *Phys. Rev. B* **96**, 085113 (2017).”
5. Page 5, Paragraph 1, Line 5-7, one sentence added to introduce the fundamental difference between BP and the binary SnSe system induced by interlayer ferroelectric dipole alignment, “However, due to the high polarizability of the binary lattice, the SnSe family forms distinct anti-ferroelectric stacking order along the interlayer a axis, when two neighbouring MLs reverse the in-plane dipole orientation along the c -axis [18,19].”
6. Page 6, Paragraph 2, Line 5-6, one sentence added to clarify the energy diagram, “Density functional theory (DFT) calculations allow us to determine the VB maximums of bulk SnSe and SnSe₂, which are 4.68 and 5.05 eV, respectively.”
7. Page 6, Paragraph 2, Line 8-13, and Page 7, Paragraph 1, Line 1-4, discussion added regarding the experimental evidence of the interfacial charge transfer mechanism, “This charge transfer controlled hole doping mechanism is verified by scanning Kelvin probe microscopy, which directly measures the surface E_F of p-SnSe samples with an energy resolution better than 50 meV. For BR1-SnSe, the deduced E_F for bare SnSe surface and few-layer SnSe₂-SnSe heterostructure on the same sample are 4.42 and 4.58 eV, respectively (Supplementary Fig. 7). Compared to intrinsic semiconducting SnSe ($E_F=4.23$ eV), the Fermi energy of BR1-SnSe crystals is shifted down by 0.19 eV, corresponding to p-type doping in the BR1-SnSe bulk. For few-layer SnSe₂-SnSe heterostructure, the Fermi energy (4.58 eV) is shifted up from the bulk value of 4.85 eV, providing an unambiguous evidence of electron transfer-in from SnSe (Supplementary Fig. 7).”
8. Page 8, Paragraph 2, Line 7-9, one sentence added to highlight the correlation between high electrical conductivity in p-SnSe and the pudding mold VB1, “Such X-shaped VB1 also explains the experimental observation of relatively high electrical conductivity in p-SnSe, as a result of small effective mass (m_e) and multi-valley contributions to charge transport.”
9. Page 9, Paragraph 2, Line 7-9, one sentence added to explain the intervalley scattering mechanism, “Consequently, intervalley scattering is expected to be stronger in the case of SF1-SnSe, when the momentum mismatch Δp is compensated by the dipole-field acceleration of hole carriers in the case of $I \parallel c$.”

Reviewers' Comments:

Reviewer #1:

Remarks to the Author:

The authors have thoroughly responded to my comments as well as to comments by the other two referees. They added appropriate lines and figures/insets in both the main text and the Supporting Information.

In my opinion, this is now a very very interesting paper on an important topic and I expect the paper to generate numerous citations.

My recommendation is publish as is.

Reviewer #2:

Remarks to the Author:

During the revised stage, authors addressed all the comments that I have put, it can be published as is. One minor point should be revised:

In the second paragraph of page 7, when authors mentioned that the carrier concentration of SF3 sample is comparable to that in ref.2, I suggest to add discussion: present results indicate that the reported results from ref 2 is one possible case with a given cooling rate, as well elucidated in this work. Therefore, we believe that the not only the prepared methods, even also one of the processing parameters (such as cooling rate) will significantly affect the thermoelectric transport properties, which is associated with crystal defects (one work should be cited here, Wu et al. Nano Energy, 35 (2017) 321-330).

Reviewer #3:

Remarks to the Author:

I appreciate authors' efforts to address the main issues raised by the three reviewers. The responses and revisions are satisfactory. I would thus recommend the manuscript for publication in as is.

II. Replies to the Reviewers' comments

Reviewer #1:

Comment: The authors have thoroughly responded to my comments as well as to comments by the other two referees. They added appropriate lines and figures/insets in both the main text and the Supporting Information.

In my opinion, this is now a very very interesting paper on an important topic and I expect the paper to generate numerous citations.

My recommendation is publish as is.

Reply: We are thankful to the Referee for his/her recognition of the importance of our work.

Reviewer #2:

Comment: *During* the revised stage, authors addressed all the comments that I have put, it can be published as is. One minor point should be revised:

In the second paragraph of page 7, when authors mentioned that the carrier concentration of SF3 sample is comparable to that in ref.2, I suggest to add discussion: present results indicate that the reported results from ref 2 is one possible case with a given cooling rate, as well elucidated in this work. Therefore, we believe that the not only the prepared methods, even also one of the processing parameters (such as cooling rate) will significantly affect the thermoelectric transport properties, which is associated with crystal defects (one work should be cited here, Wu et al. Nano Energy, 35 (2017) 321-330).

Reply: The referee's suggestion is really important by pointing out the critical importance of specific crystal-growth processing parameters (here, it is specifically referring to the flux cooling rate).

In response, we have incorporated his suggestion into the main text of the revised manuscript., See Page 7, Paragraph 2, Line 6-13, "Our results strongly suggest the critical importance of the flux cooling rate in determining the thermoelectric transport properties of SnSe, by controlling the concentration of SnSe₂ micro-domains in the bulk. This is mainly due to the fact the nucleation of SnSe and SnSe₂ are rather close in temperatures, which are ~ 820 °C and ~ 650 °C, respectively. For SnSe crystal with very fast flux cooling rates, Sn vacancies and Se interstitials, which may effectively scatter lattice phonons, have been imaged by scanning transmission electron microscope [23]. However, these two specific types of defects are not present in our SnSe single crystals."

Note that it is appropriate to compare our single crystals with samples in the aforementioned paper [Wu et al. Nano Energy, 35 (2017) 321-330], since they are completely different in the nature of defects. However, we thank the Referee for reminding us the existence of this interesting paper, and we have cited it with proper discussions.

Reviewer #3:

Comment: I appreciate authors' efforts to address the main issues raised by the three reviewers. The responses and revisions are satisfactory. I would thus recommend the manuscript for publication in as is.

Reply: We thank the Referee for the recommendation.